# *L2HGDH* Missense Variant in a Cat with L-2-Hydroxyglutaric Aciduria

**DOI:** 10.3390/genes12050682

**Published:** 2021-05-01

**Authors:** Matthias Christen, Nils Janzen, Anne Fraser, Adrian C. Sewell, Vidhya Jagannathan, Julien Guevar, Tosso Leeb, Daniel Sanchez-Masian

**Affiliations:** 1Institute of Genetics, Vetsuisse Faculty, University of Bern, 3001 Bern, Switzerland; matthias.christen@vetsuisse.unibe.ch (M.C.); vidhya.jagannathan@vetsuisse.unibe.ch (V.J.); 2Screening Labor Hannover, 30430 Hannover, Germany; n.janzen@metabscreen.de; 3Department of Clinical Chemistry, Hannover Medical School, 30625 Hannover, Germany; 4Anderson Moores Veterinary Specialists, Winchester SO21 2LL, Hampshire, UK; Anne.Fraser@andersonmoores.com (A.F.); dsanchezmasian@gmail.com (D.S.-M.); 5Biocontrol, 55128 Ingelheim, Germany; acsewell1950@gmail.com; 6Department of Clinical Veterinary Medicine, Vetsuisse Faculty, University of Bern, 3001 Bern, Switzerland; julien.guevar@vetsuisse.unibe.ch

**Keywords:** *Felis catus*, animal model, neurology, metabolism, metabolite repair, seizure, precision medicine

## Abstract

A 7-month-old, spayed female, domestic longhair cat with L-2-hydroxyglutaric aciduria (L-2-HGA) was investigated. The aim of this study was to investigate the clinical signs, metabolic changes and underlying genetic defect. The owner of the cat reported a 4-month history of multiple paroxysmal seizure-like episodes, characterized by running around the house, often in circles, with abnormal behavior, bumping into obstacles, salivating and often urinating. The episodes were followed by a period of disorientation and inappetence. Neurological examination revealed an absent bilateral menace response. Routine blood work revealed mild microcytic anemia but biochemistry, ammonia, lactate and pre- and post-prandial bile acids were unremarkable. MRI of the brain identified multifocal, bilaterally symmetrical and T2-weighted hyperintensities within the prosencephalon, mesencephalon and metencephalon, primarily affecting the grey matter. Urinary organic acids identified highly increased levels of L-2-hydroxyglutaric acid. The cat was treated with the anticonvulsants levetiracetam and phenobarbitone and has been seizure-free for 16 months. We sequenced the genome of the affected cat and compared the data to 48 control genomes. *L2HGDH*, coding for L-2-hydroxyglutarate dehydrogenase, was investigated as the top functional candidate gene. This search revealed a single private protein-changing variant in the affected cat. The identified homozygous variant, XM_023255678.1:c.1301A>G, is predicted to result in an amino acid change in the L2HGDH protein, XP_023111446.1:p.His434Arg. The available clinical and biochemical data together with current knowledge about *L2HGDH* variants and their functional impact in humans and dogs allow us to classify the p.His434Arg variant as a causative variant for the observed neurological signs in this cat.

## 1. Introduction

L-2-hydroxyglutaric aciduria (L-2-HGA) is a rare metabolic disorder that was initially reported in a five-year-old boy with psychomotor retardation and musculoskeletal dystrophy [1] (OMIM #236792). L-2-HGA has since also been described in different dog breeds [2,3,4]. Clinical and radiological signs were reported in a cat with 2-HGA, but the causative defect at the molecular level was not reported [5]. Clinical signs of patients in veterinary medicine typically include gait dysfunction, behavioral changes and seizure-like/dyskinetic episodes [6]. The first clinical signs in affected dogs become apparent at a mean age of twelve months, and both sexes seem to be affected equally [6].

The main diagnostic criteria for L-2-HGA in all species are the marked elevation of L-2-hydroxyglutaric acid in urine or cerebrospinal fluid [1], as well as pathognomonic MRI changes [2,7]. After the first description of disease-causing variants in the *L2HGDH* gene, coding for the enzyme L-2-hydroxyglutarate dehydrogenase [8], genetic investigation of suspected cases has been used as an additional means to confirm putative diagnosis. Disease-causing variants for this autosomal recessive inherited disease have been reported in humans [8,9] and dogs [10,11] (OMIA 001371-9615).

Different murine models for the disease have been introduced over the last years in order to further our understanding of the affected metabolic pathway [12,13]. L-2-hydroxyglutarate is produced as the result of the reduction of α-ketoglutarate during the Krebs cycle. This reduction is mediated through unspecific side reactions of L-malate dehydrogenase [12,14] and L-lactate dehydrogenase [15]. Under normoxic conditions, L-2-hydroxyglutarate does not possess any known physiological function in eukaryotes [14]. During hypoxic conditions, the synthesis of L-2-hydroxyglutarate is increased, and it is involved in regulating the adaptation to hypoxia [15,16]. The FAD-dependent metabolite repair enzyme L-2-hydroxyglutarate dehydrogenase is needed to facilitate its oxidation back to α-ketoglutarate [14,17]. If this mechanism fails, L-2-hydroxyglutarate is accumulated in different tissues. In particular, it is pumped into astrocytes [18]. The changes in astrocytes, together with neuroinflammation of microglia, are part of the gliosis seen in mouse models for L-2-HGA [13]. As L-2-hydroxyglutarate is a structural analog of α-ketoglutarate, it is additionally thought to inhibit enzymes that use α-ketoglutarate as a substrate [12,19,20,21].

The cause of L-2-HGA remains obscure. There is experimental evidence that intracerebral administration of L-2-hydroxyglutaric acid to neonatal rats induces reactive oxygen species leading to cerebral vacuolation and edema [22], confirming a previous observation of apoptosis in a mouse model [13]. A recent report of L-2-hydroxyglutaric acid inducing mitochondrial stress and neuronal dysfunction [23] appears to support an effect on redox homeostasis in the pathogenesis of L-2-HGA.

This study was initiated after the presentation of a 7-month-old, female spayed, domestic longhair cat with neurological episodes and MRI findings resembling L-2-HGA. The goal of the study was to characterize the clinical, radiological and metabolic phenotype and to investigate a possible underlying causative genetic defect.

## 2. Materials and Methods

### 2.1. Ethics Statement

All animal experiments were performed according to local regulations. The cat in this study is privately owned and was examined with the consent of the owner. The Cantonal Committee for Animal Experiments approved the collection of blood samples from control cats that were used in this study (Canton of Bern; permit BE 71/19).

### 2.2. Clinical Examination

A 7-month old, female spayed, domestic longhair cat was investigated. The clinical, neurological and radiologic examinations were done by board-certified veterinary neurologists (D.S-M., A.F). Lithium heparin blood samples were collected for serum biochemistry, ammonia, lactate and bile acids stimulation test. EDTA blood samples were collected for hematology and genomic DNA isolation. A urine sample was collected by cystocentesis under general anesthesia for metabolic screening after the MRI-scan.

### 2.3. Magnetic Resonance Imaging

An MRI study of the brain using a 1.5 T magnet (Achieva, Philips Medical Systems, Guildford, UK) was performed with the cat under general anesthesia. The cat was sedated with intravenous dexmedetomidine (Dexdomitor^®^) and butorphanol (Torbutrol^®^), and general anesthesia was induced with an intravenous injection of Alfaxalone (Alfaxan^®^). The cat was intubated, and anesthesia was maintained with a mixture of Sevoflurane (SevoFlo^®^—Zoetis) and oxygen. Dorsal T2-weighted (T2W), sagittal T2W and transverse T1-weighted (T1W), T2W, fluid attenuating inversion recovery (FLAIR), T2*-weighted (T2*W), diffusion-weighted images (DWI) and apparent diffusion coefficient (ADC) images were acquired. Transverse T1W and three-dimensional (3D) T1W turbo field echo series with a voxel size of 1 × 1 × 1 mm were also acquired after intravenous administration of 0.1 mmol/kg gadopentetate dimeglumine (Magnevist, Bayer Schering Pharma AG, Berlin, Germany).

### 2.4. Metabolic Screening

Based on the MRI findings, an EDTA blood sample and a urine sample collected by cystocentesis were submitted for amino acid and organic acid screening as a neurometabolic disease was suspected.

The EDTA whole blood was spotted onto a filter card commonly used for newborn screening and dried for 4 h at room temperature. One 3.2 mm diameter blood spot was then punched out. One hundred microliters methanol: water (80:20 by volume) and deuterated acylcarnitine standards (MassChrom^®^ amino acids and acylcarnitines from dried blood, Chromsystems, Munich, Germany) were added, and the spot was eluted for 30 min. Free and total carnitine, acylcarnitines and amino acids were determined and quantified by FIA-MS/MS (Waters, Eschborn, Germany). The NeoLynx software was used for evaluation (Waters, Eschborn, Germany).

As a pre-test, a part of the urine sample was oximated, the organic acids extracted, derivatized into trimethylsilyl compounds and separated by qualitative GC/MS (Shimadzu, Duisburg, Germany) according to the method described by Sweetman et al. [24].

Differentiation of the L- and D-2-hydroxyglutarate enantiomers was performed as described by Struys et al. [25]. Briefly, urine was diluted with methanol and spiked with the internal standard (deuterated D-/L-2-hydroxyglutarate) and then dried to dryness under nitrogen. The residue was then added to 50 µL of freshly prepared diacetyl-L-tartrate anhydride in dichloromethane: acetic acid (4:1 by volume) and incubated for 30 min at 75 °C. After subsequent cooling and re-evaporation under a nitrogen stream, the residue was taken up in 0.5 mL of LC-pure water and 10 µL were injected. Separation was performed from a C18 Xterra C18 analytical 150/3.9 mm column and gradient elution with water–acetonitrile (96.5:3.5 by volume) containing 125 mg/L ammonium formate (pH adjusted to 3.6) on a Waters LC–MS/MS ZZZ (Waters, Eschborn, Germany). Detection was performed in negative MRM mode and quantification using the internal standard and the QuanLynx software (Waters, Eschborn, Germany).

The concentration of L-2-hydroxgluarate was related to the urinary creatinine value. This was determined by a kinetic Jaffe method as part of the routine clinical chemistry on a COBAS 6000 analyzer (Roche Diagnostics, Mannheim).

### 2.5. Control Samples for Genetic Analyses

In addition to the affected cat, 582 blood samples from cats of different breeds, which had been donated to the Vetsuisse Biobank, were used. They represented unrelated population controls without reports of a similar neurological phenotype.

### 2.6. DNA Extraction

Genomic DNA was isolated from EDTA blood with the Maxwell RSC Whole Blood Kit using a Maxwell RSC instrument (Promega, Dübendorf, Switzerland).

### 2.7. Whole-Genome Sequencing

An Illumina TruSeq PCR-free DNA library with ~500 bp insert size of the affected cat was prepared. We collected 146 million 2 × 150 bp paired-end reads on a NovaSeq 6000 instrument (14.5x coverage). Mapping to the Felis_catus_9.0 reference genome assembly was performed as described [26]. The sequence data were deposited under the study accession PRJEB7403 and the sample accession SAMEA7376283 at the European Nucleotide Archive. Genome sequence data of 48 control cats were also included in the analysis (Appendix A).

### 2.8. Variant Calling

Variant calling was performed using GATK HaplotypeCaller [27] in gVCF mode as described [23]. To predict the functional effects of the called variants, SnpEff [28] software, together with the Felis_catus_9.0 reference genome assembly and NCBI annotation release 104, was used.

### 2.9. Gene Analysis

Numbering within the feline *L2HGDH* gene corresponds to the NCBI RefSeq accession numbers XM_023255678.1 (mRNA) and XP_023111446 (protein).

### 2.10. In Silico Functional Predictions and Database Searches

PredictSNP [29], PROVEAN [30], and MutPred2 [31] were used to predict biological consequences of the discovered protein variant. The feline and human L2HGDH proteins both comprise 463 amino acids, of which 407 (88%) are identical between cat and human. The Genome Aggregation Database (gnomAD) [32] and Online Mendelian Inheritance in Animals database [33] were searched for corresponding variants in the human and animal *L2HGDH* genes.

### 2.11. PCR and Sanger Sequencing

Primers 5′-GCC TTT GGC TAA ACA CAA ACC T -3′ (Primer F) and 5′- GCT GGT GAG GAG AGG TCC AT -3′ (Primer R) were used for the generation of an amplicon containing the *L2HGDH*:c.1301A>G variant. PCR products were amplified from genomic DNA using AmpliTaq Gold 360 Master Mix (Thermo Fisher Scientific, Reinach, Switzerland). Direct Sanger sequencing of the PCR amplicons on an ABI 3730 DNA Analyzer (Thermo Fisher Scientific, Reinach, Switzerland) was performed after treatment with exonuclease I and alkaline phosphatase. Sanger sequences were analyzed using the Sequencher 5.1 software (Gene Codes, Ann Arbor, MI, USA).

## 3. Results

### 3.1. Clinical History and Examination

The cat presented with a 4-month history of approximately weekly paroxysmal seizure-like episodes, characterized by abnormal behavior, running uncontrollably around the house, often in circles, bumping into obstacles, salivation and often urination. The episodes were approximately 30–60 s in duration followed by a period, generally one day in duration, of being distressed, hiding from the owner and being inappetent. Between the seizure-like episodes, the cat was reported normal other than having occasional hallucination-like episodes characterized by staring fixedly and pawing at the air. General physical examination was unremarkable. No abnormalities were noted on neurological examination apart from an absent menace response bilaterally.

### 3.2. Radiological Examination

MRI of the brain showed multifocal, bilaterally symmetrical, T2W hyperintensities, which were iso- to hyperintense on FLAIR, primarily affecting the grey matter of the diencephalon, mesencephalon and metencephalon, with no associated contrast enhancement or mass effect (Figure 1). The T2W and FLAIR hyperintensities were T1W iso- to hypo-intense and affected the thalamus, hypothalamus, red nuclei, the region of the periaqueductal grey matter, rostral and caudal colliculi, the region of the rostral and middle cerebellar peduncles, the confluence of the peduncles, the region of the vestibular nuclei, the cerebellar nuclei (lateral, interposital and fastigial), and the cerebellar grey matter (in particular the vermis) diffusely. Additional bilaterally symmetrical T2W and FLAIR hyperintensities were identified at the grey-white matter junction of the telencephalon. The cerebral cortex showed diffuse, mildly increased signal intensity on T2W and FLAIR images. The cerebral sulci were prominent, and there was mild enlargement of the lateral and third ventricles.

### 3.3. Laboratory Findings

Hematological examination, including smear evaluation, identified a mild microcytic anemia, while white blood cell morphology was otherwise unremarkable. Complete serum biochemistry analysis was unremarkable. Pre- and post-prandial bile acids were unremarkable, as well as blood ammonia and lactate concentrations.

In the GC-MS profile of organic acids in urine, several peaks were found that could be attributed to drugs, e.g., levetiracetam, as well as evidence for 2-hydroxyglutaryllactone. In addition, a peak of 2-hydroxyglutaric acid could be clearly detected. No or only traces of this compound were found in the urine of unaffected subjects. Separation of the enantiomers of 2-hydroxyglutaric acid in the affected cat showed a clearly increased excretion of L-2-hydroxyglutarate (99.1%) compared to the D-form (0.9%). The enantiomeric analysis confirmed the L-configuration and established the diagnosis of L-2-HGA.

Decreased free carnitine of 2.96 µmol/L (ref.: 5.0–33.8 µmol/L) and a relatively low total carnitine of 6.0 µmol/L (ref.: 5.2–44.4 µmol/L) were found in blood [34]. This is compatible with a (secondary) carnitine deficiency. All acylcarnitines and amino acids measured were unremarkable relative to feline or human reference values.

### 3.4. Genetic Analysis

As clinical, radiological and biochemical findings resembled previously published cases of companion animals with L-2-HGA [4,6], we hypothesized that the phenotype in the affected cat was due to a variant in the *L2HGDH* gene. Hence, *L2HGDH* was investigated as the top functional candidate gene. We sequenced the genome of the affected cat and searched for private homozygous and heterozygous variants that were not present in the genome sequences of 48 control cats (Table 1 and Appendix A).

This analysis identified a single homozygous private protein-changing variant in *L2HGDH*. This missense variant, XM_023255678.1:c.1301A>G, is predicted to result in an amino acid substitution in the L2HGDH protein, XP_023111446.1:p.His434Arg.

The histidine-to-arginine substitution was predicted pathogenic and deleterious by several in silico prediction tools (PredictSNP probability for pathogenicity: 87%; MutPred2 score: 0.878; PROVEAN score: −7.729).

We confirmed the presence of the *L2HGDH* variant in a homozygous state in the affected cat by Sanger sequencing (Figure 2). We also genotyped the variant in 534 additional control cats of different breeds. None of these cats carried the mutant allele.

### 3.5. Treatment and Outcome

Treatment with anticonvulsant drug levetiracetam 20 mg/kg three times per day was initiated, and an initial reduction in seizure frequency was noted. The seizure frequency subsequently increased over a couple of months, and treatment with anticonvulsant drug phenobarbitone 2 mg/kg twice daily was then initiated. Levetiracetam was reduced to twice daily administration due to side effects (sedation). Following the initiation of phenobarbitone therapy, the cat has remained seizure-free for 16 months, although it has continued to have occasional hallucination-like episodes.

## 4. Discussion

In this study, we identified a homozygous *L2HGDH*:c.1301A>G missense variant in a domestic cat with a history of suspected seizure-like episodes, corresponding MRI-changes and markedly elevated levels of L-2-hydroxyglutaric acid in the obtained urine sample. The combination of the MRI abnormalities (i.e., multifocal, bilaterally symmetrical, T2W hyperintensities, primarily affecting the grey matter in the mesencephalon and metencephalon) appears to be a common feature for L-2-HGA in cats [5]. Epileptic seizure activity has also been reported in a cat previously with 2-HGA; however, the phenotype of the epileptic seizure activity was different. In the previous case [5], generalized tonic-clonic epileptic seizures were reported, while our case displayed multiple episodes of running frantically around the house, often in circles, for approximately 30–60 s, bumping into obstacles, having altered mentation and autonomic signs, including salivation and often urination. These episodes are similar to those described in some dogs with L-2-HGA [6]. Despite the lack of a specific treatment for this neurometabolic disease, anti-epileptic medication seems to improve the frequency of the epileptic seizure activity. *L2HGDH* encodes the metabolite repair enzyme L-2 hydroxyglutarate dehydrogenase. This enzyme mediates the oxidation of L-2 hydroxyglutarate, the product of a weak side-reaction of L-malate dehydrogenase [12,14] and L-lactate dehydrogenase [15] during the Krebs cycle, to α-ketoglutarate [14,16].

Variants in human and canine *L2HGDH* cause L-2-HGA [6,9]. Interestingly, the variant described herein constitutes an amino acid change at the same position as one of the two amino acid changes discovered in Staffordshire Bull Terriers by Penderis et al. (XP_038529514.1:Leu433Pro, His434Tyr, OMIA 001371-9615) [10]. The authors suggested that the two variants could have occurred separately with only one being causative. Our data in the cat suggest that the substitution of ^434^His alone is sufficient to disrupt the function of the L2HGDH enzyme.

Unfortunately, we did not have access to the parents of the affected cat or any other heterozygous cat. It would have been interesting to see if heterozygous cats show altered values of organic acids in urine or an increased susceptibility to seizures or other neurologic signs.

Recently, another cat with similar clinical and biochemical findings was also diagnosed with 2-HGA. In this cat, the causative genetic variant was not investigated [5]. As this cat also originated in the United Kingdom, it seems remotely possible that they shared the same genetic defect and that the disease allele is present at non-negligible frequency in the local cat population.

Screening for this condition on the biochemical and/or genetic level may help to distinguish between cats suffering from L-2-HGA and other varieties of epileptic seizure disorders, thus potentially enabling specific treatment for patients.

## 5. Conclusions

We identified a domestic cat with L-2-HGA that clinically and genetically resembled human and canine patients with L-2-HGA. To the best of our knowledge, this cat represents the first feline patient with a known spontaneous disease-causing variant in the *L2HGDH* gene. This study demonstrated mutual corroboration of the biochemical and genetic investigations.

## Figures and Tables

**Figure 1 genes-12-00682-f001:**
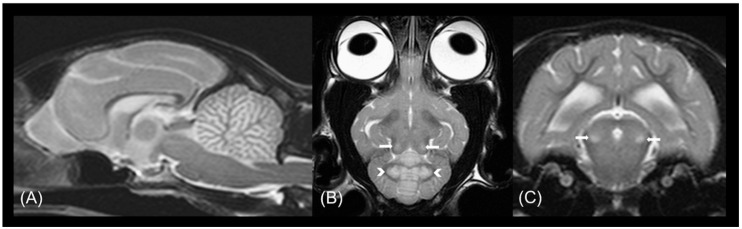
MRI abnormalities in a cat with L2HGA. (**A**) Mid-sagittal T2-weighted, (**B**) dorsal T2-weighted and (**C**) transverse T2-weighted MR images demonstrating symmetrical involvement of predominantly grey matter structures are shown. Consistent bilateral symmetrical T2 hyperintensity of the deep cerebellar nuclei (white arrowheads) and caudal colliculi (white arrows) and diffuse T2-weighted hyperintensity affecting the cerebellar cortex can be seen.

**Figure 2 genes-12-00682-f002:**
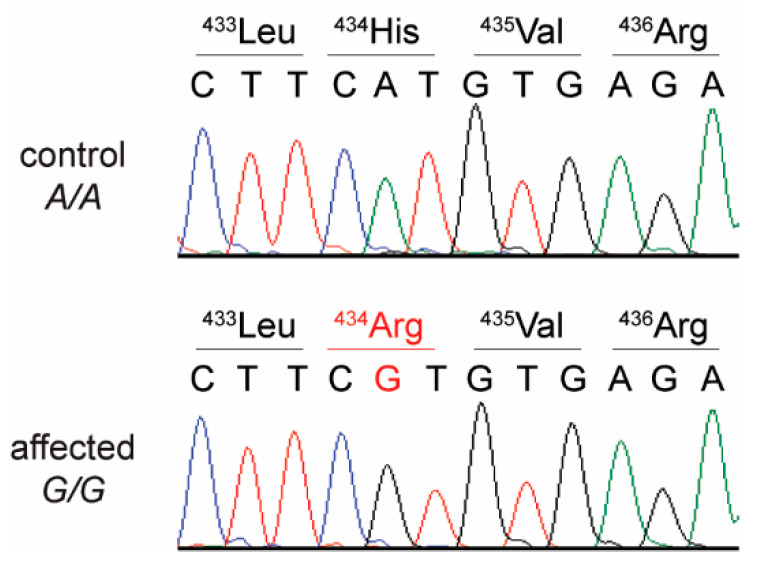
Details of the *L2HGDH*:c.c.1301A>G variant. Representative electropherograms of a control and the affected cat are shown. The amino acid translations of the wild type and mutant alleles are indicated.

**Table 1 genes-12-00682-t001:** Results of variant filtering in the affected cat against 48 control genomes.

Filtering Step	Homozygous Variants	Heterozygous Variants
All variants in the affected cat	4,038,732	4,552,718
Private variants	11,860	69,034
Protein-changing private variants	61	336
Private variants in *L2HGDH* candidate gene	1	0

## Data Availability

The accessions for the sequence data reported in this study are listed in Appendix A.

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
