# Peer review of "L2HGDH Missense Variant in a Cat with L-2-Hydroxyglutaric Aciduria"

_genes, 2021, doi:10.3390/genes12050682_

Round 1

Reviewer 1 Report

This paper presents the first full genetic and biochemical characterisation of L-2-hydroxyglutaric aciduria in a cat.

The cat had presented with neurological signs and MRI findings that were considered pathognomonic with elevated  L-2-hydroxyglutarate.

The authors have sequenced the affected cat's genome and searched for private homozygous and heterozygous variants that were not present in the genome sequences of 48 control cats. The analysis identified a single homozygous private protein-changing variant, predicted to result in an amino acid substitution in the L2HGDH protein.

Major comments

In the introduction, lines 60-62 make a statement that L-2-hydroxyglutarate does not possess any known physiological function, quoting a statement in a 2009 review. However, more recent data show a role for L-2-hydroxyglutarate as a molecule that can mediate physiological responses in hypoxia.

See:

  • Intlekofer AM, et al. Hypoxia Induces Production of L-2-Hydroxyglutarate. Cell Metab. 2015;22:304–11
  • Oldham WM, Clish CB, Yang Y, Loscalzo J. Hypoxia-Mediated Increases in L-2-hydroxyglutarate Coordinate the Metabolic Response to Reductive Stress. Cell Metab. 2015;22:291–303.
  • Hunt, R. J., Granat, L., McElroy, G. S., Ranganathan, R., Chandel, N. S., & Bateman, J. M. (2019). Mitochondrial stress causes neuronal dysfunction via an ATF4-dependent increase in L-2-hydroxyglutarate. The Journal of Cell Biology, 218(12), 4007-4016. https://doi.org/10.1083/jcb.201904148

This may be important for greater understanding in the future regarding how or why neurological episodes could be triggered in the affected animal or person, so I suggest that the introduction be adapted to include these or other appropriate papers.

Minor comments

Line 44-45 "Similar clinical and radiological signs..." As the clinical signs etc have not been described at this point in the paper, this sentence is unhelpful. Perhaps state the signs reported in the Nye et al. paper.

Line 76 …approved the collection of blood samples in all cat samples used in this study (Canton of Bern,….

Line 79  replace “were done by a board-certified veterinary neurologists” by “were performed by board-certified veterinary neurologists”.

Line 81-82 Clarify whether the blood sample and cystocentesis were carried under sedation/anaesthesia- ie while the cat was under the same period of anaesthesia as for the MRI.

Line 85- state the anaesthetic used.

It is not clear regarding the decision-making sequence. Where was that moment where the neurologists recognised that this presentation might be L-2-HGA? The introduction suggests that it was following the MRI. The materials and methods and subsequent results do not clarify the sequence of events. This sequencing may also help to explain why control bloods from other cats were not submitted simultaneously for amino acid and organic acid screening to help establish laboratory reference ranges- see note below.

Line 186- “increase” implies a baseline normal- what were the values and reference range?

Line 189-190. Human reference values are given here for serum/plasma carnitine. There are feline reference values available, and these should be referenced in this publication.

i.e. Fischer, J. (1993). L-carnitine plasma levels in dogs and cats A diagnostic parameter?. Bibliographic information available from INIS: http://inis.iaea.org/search/search.aspx?orig_q=RN:25025504; Available from the Veterinaermedizinische Univ. Bibliothek, Linke Bahngasse 11, 1030 Vienna (AT)

This thesis's abstract states that normal values of free carnitine in cats are 8.2 to 24.2 μmol/l, which is somewhat lower than the human reference ranges given here as 20-70 μmol/L. Other papers also indicate similar ranges for feline plasma total carnitine.

Line 196-197 - suggest that the authors move this sentence upwards to the other urine results- i.e. after line 188.

Line 288- as the urine sample was obtained by cystocentesis, should this not say “in the obtained urine sample” and not “submitted”, as “submitted” implies that the owner obtained it?

Line 249- remove the extra comma after “see”.

Author Response

(1)

In the introduction, lines 60-62 make a statement that L-2-hydroxyglutarate does not possess any known physiological function, quoting a statement in a 2009 review. However, more recent data show a role for L-2-hydroxyglutarate as a molecule that can mediate physiological responses in hypoxia.

See:

Intlekofer AM, et al. Hypoxia Induces Production of L-2-Hydroxyglutarate. Cell Metab. 2015;22:304–11

Oldham WM, Clish CB, Yang Y, Loscalzo J. Hypoxia-Mediated Increases in L-2-hydroxyglutarate Coordinate the Metabolic Response to Reductive Stress. Cell Metab. 2015;22:291–303.

Hunt, R. J., Granat, L., McElroy, G. S., Ranganathan, R., Chandel, N. S., & Bateman, J. M. (2019). Mitochondrial stress causes neuronal dysfunction via an ATF4-dependent increase in L-2-hydroxyglutarate. The Journal of Cell Biology, 218(12), 4007-4016. https://doi.org/10.1083/jcb.201904148

This may be important for greater understanding in the future regarding how or why neurological episodes could be triggered in the affected animal or person, so I suggest that the introduction be adapted to include these or other appropriate papers.

Response: We thank the reviewer for pointing out these publications to us. We revised the introduction accordingly.

(2)

Line 44-45 "Similar clinical and radiological signs..." As the clinical signs etc have not been described at this point in the paper, this sentence is unhelpful. Perhaps state the signs reported in the Nye et al. paper.

Response: We revised the sentence.

(3)

Line 76 …approved the collection of blood samples in all cat samples used in this study (Canton of Bern,….

Response: We slightly modified the ethics statement following the reviewer’s suggestion. Our ethics permit is valid for Switzerland, where the control cats were samples. The blood from the affected cat was leftover material from a sample taken for diagnostic purposes and did not require an ethics permit.

(4)

Line 79  replace “were done by a board-certified veterinary neurologists” by “were performed by board-certified veterinary neurologists”.

Response: Revised accordingly.

(5)

Line 81-82 Clarify whether the blood sample and cystocentesis were carried under sedation/anaesthesia- ie while the cat was under the same period of anaesthesia as for the MRI.

Response: Revised accordingly.

(6)

Line 85- state the anaesthetic used.

Response: Revised accordingly.

(7)

It is not clear regarding the decision-making sequence. Where was that moment where the neurologists recognised that this presentation might be L-2-HGA? The introduction suggests that it was following the MRI. The materials and methods and subsequent results do not clarify the sequence of events. This sequencing may also help to explain why control bloods from other cats were not submitted simultaneously for amino acid and organic acid screening to help establish laboratory reference ranges- see note below.

Response: Indeed, the suspicion that this might represent L-2-HGA arose after the MRI. We slightly revised the methods section to make this more clear.

(8)

Line 186- “increase” implies a baseline normal- what were the values and reference range?

Response: We rephrased this statement, which referred to a semi-quantitative observation.

(9)

Line 189-190. Human reference values are given here for serum/plasma carnitine. There are feline reference values available, and these should be referenced in this publication.

i.e. Fischer, J. (1993). L-carnitine plasma levels in dogs and cats A diagnostic parameter?. Bibliographic information available from INIS: http://inis.iaea.org/search/search.aspx?orig_q=RN:25025504; Available from the Veterinaermedizinische Univ. Bibliothek, Linke Bahngasse 11, 1030 Vienna (AT)

This thesis's abstract states that normal values of free carnitine in cats are 8.2 to 24.2 μmol/l, which is somewhat lower than the human reference ranges given here as 20-70 μmol/L. Other papers also indicate similar ranges for feline plasma total carnitine.

Response: We found a more recent reference for feline reference values of free carnitine and carnitine (corresponding to total carnitine) [Côté, 2015, ref.34]. In ref. [34], free carnitine is reported to be 3.9-18.3 µmol/L (males) and 5.0-33.8 µmol/L (females). Total carnitine reference values are 5.0-22.2 µmol/L (males) and 5.2-44.4 µmol/L (females). The information in the text is now supplemented with the feline reference values and reference [34] is cited. We removed the human reference values.

(10)

Line 196-197 - suggest that the authors move this sentence upwards to the other urine results- i.e. after line 188.

Response: Revised accordingly.

(11)

Line 288- as the urine sample was obtained by cystocentesis, should this not say “in the obtained urine sample” and not “submitted”, as “submitted” implies that the owner obtained it?

Response: Revised accordingly.

(12)

Line 249- remove the extra comma after “see”.

Response: Revised accordingly.

Reviewer 2 Report

L-2-hydroxyglutaric aciduria (L-2-HGA) is a rare metabolic disorder that was reported in humans end animals. The authors made an attempt to study this phenomenon in the cat. This study demonstrated  mutual corroboration of the biochemical and genetic investigations.

Revision

1. I recommend redrafting the summary and clearly specifying the aim.
2. The conclusion in abstract is rather more speculative.
3. The authors used "PredictSNP[26],   PROVEAN [27],   and   MutPred2 [28] to predict the biological consequences of discovered protein variant". To clearly predict the "biological function/consequences" of protein, only the assays based on protein analyzes should be performed, such as WB, to assess the presence or absence of protein products/variants

Author Response

(1)

I recommend redrafting the summary and clearly specifying the aim.

Response: We revised the abstract and added the aim of the investigation.

(2)

The conclusion in abstract is rather more speculative.

Response: We thank the reviewer for this comment. Indeed, the biochemical analysis definitively confirmed the suspected genetic defect. We now phrased the conclusion in the abstract less speculative.

(3)

The authors used "PredictSNP[26],   PROVEAN [27],   and   MutPred2 [28] to predict the biological consequences of discovered protein variant". To clearly predict the "biological function/consequences" of protein, only the assays based on protein analyzes should be performed, such as WB, to assess the presence or absence of protein products/variants.

Response: The important experiment is the biochemical analysis of the metabolites. The massive increase in L-2-hydroxyglutarate proves the functional defect in L-2 hydroxyglutarate dehydrogenase (L2HGDH) as this is the only enzyme that metabolizes this substrate. A Western blot as suggested by the reviewer is unlikely to show any difference as the missense variant will not lead to a detectable change in the size of the expressed mutant L2HGDH protein. We did not change the manuscript with regard to this comment.